# A Life Course Approach to the Prevention of Iron Deficiency Anemia in Indonesia

**DOI:** 10.3390/nu14020277

**Published:** 2022-01-10

**Authors:** Ali Sungkar, Saptawati Bardosono, Rima Irwinda, Nurul R. M. Manikam, Rini Sekartini, Bernie E. Medise, Sri S. Nasar, Siti Helmyati, Anna Surti Ariani, Juntika Nurihsan, Emi Nurjasmi, Levina Chandra Khoe, Charisma Dilantika, Ray Wagiu Basrowi, Yvan Vandenplas

**Affiliations:** 1Department of Obstetric Gynecology, Medical Faculty, Universitas Indonesia—Dr. Cipto Mangunkusumo Hospital, Central Jakarta 10430, Indonesia; alisungkar03@gmail.com (A.S.); rima.irwinda@yahoo.com (R.I.); 2Department of Nutrition, Medical Faculty, Universitas Indonesia—Dr. Cipto Mangunkusumo Hospital, Jakarta 10430, Indonesia; tati.bardo@yahoo.com (S.B.); nurul.ratna@hotmail.com (N.R.M.M.); 3Department of Pediatric, Medical Faculty, Universitas Indonesia—Dr. Cipto Mangunkusumo Hospital, Jakarta 10430, Indonesia; rsekartini@yahoo.com (R.S.); bernie.medise@yahoo.com (B.E.M.); 4Nutrition and Metabolic Disease Indonesian Pediatric Society, Jakarta Selatan 12560, Indonesia; srinasar@gmail.com; 5Department of Nutrition and Health, Faculty of Medicine, Public Health, and Nursing, Universitas Gadjah Mada, Yogyakarta 55281, Indonesia; siti.helmyati@gmail.com; 6Clinical Psychologist Indonesia, Jakarta 12190, Indonesia; ninateguh@gmail.com; 7Educational Psychology and Guidance Department, Universitas Pendidikan Indonesia, Bandung 40154, Indonesia; juntikanurihsan@upi.edu; 8Indonesia Midwives Association, Jakarta Pusat 10560, Indonesia; emitaufik@yahoo.com; 9Department of Community Medicine, Medical Faculty, Universitas Indonesia, Jakarta 10430, Indonesia; levinachandramd@gmail.com (L.C.K.); ray.basrowi@gmail.com (R.W.B.); 10Danone Specialized Nutrition, Jakarta 12950, Indonesia; charisma.dilantika@danone.com; 11KidZ Health Castle, Vrije Universiteit Brussel, UZ Brussel, 1090 Brussels, Belgium

**Keywords:** iron deficiency anemia, life course approach, nutrition

## Abstract

Iron deficiency anemia (IDA) has a long-term impact on each life stage and remains worldwide a major public health problem. Eleven experts were invited to participate in a virtual meeting to discuss the present situation and the available intervention to prevent iron deficiency anemia in Indonesia. The experts consisted of obstetric gynecologists, pediatricians, nutritionists, midwives, a clinical psychologist, and an education expert. Existing interventions focus attention on preconception and early childhood stages. Considering the inter-generational effects of IDA, we call attention to expanding strategies to all life stages through integrating political, educational, and nutritional interventions. The experts agreed that health education and nutritional intervention should be started since adolescence. Further research to explore the effectiveness of these interventions would be important for many regions in the world. The outcome of this Indonesian consensus is applicable worldwide.

## 1. Introduction

Anemia is a major public health challenge across many countries and all age groups. The most affected groups are under-five-year-old children, with a prevalence of 41.7%, followed by pregnant women (40.1%) and women of reproductive age (32.8%) [1,2,3]. In Indonesia, data from a national health survey in 2013 and 2018 showed an increasing prevalence of anemia from 37.1% to 48.9% among pregnant women and from 28% to 38.5% in under-five-year-old children [4,5]. The result from the South East Asian Nutrition Survey (SEANUTS) found an even higher prevalence of anemia, i.e., nearly 55% among children below two years old, and around 15% in children between 2 and 12 years old [6]. These age groups are the most vulnerable groups to be affected by anemia. Evidence links iron deficiency anemia with poor cognitive, motor, and social-emotional development among preschool-age children [7], while in pregnant women, anemia could result in reduced work productivity, increased risk of infections, preterm birth, poor neonatal outcome, and even maternal mortality [8,9,10,11]. Links between maternal and offspring anemia have been proposed: there is a positive correlation of the maternal hemoglobin level with the child’s iron status [12]. 

The etiology of anemia is diverse and complex. Nevertheless, iron deficiency is commonly found to be the main culprit. The World Health Organization (WHO) reported that half of the global anemia cases are due to iron deficiency [13]. Various factors are contributing to iron deficiency anemia (IDA), such as low intake of iron-rich foods, increased requirement associated with pregnancy, breastfeeding, growth spurt period in adolescence, and also low absorption in chronic diseases [14]. Poor meat, fish, or poultry availability is likely to be one of the main causes of iron deficiency in developing countries.

An iron supplementation program has widely been implemented in Indonesia, targeting female adolescents and pregnant women. However, this program phases many challenges: poor knowledge on anemia, low compliance of iron tablet consumption, low antenatal care visits, other micronutrient deficiencies, and poor health education [15,16]. Data from a basic health survey (2018) indicated that only 38.1% of pregnant women consumed the recommended iron tablets [5]. Poor knowledge of the consequences of anemia was found to be associated with poor compliance with iron tablet consumption [15]. 

Since the impact of iron deficiency is long-term and affects each life stage, it is important to consider different life stages in the prevention of IDA. 

## 2. Materials and Methods

This expert opinion was generated based on the expert meeting supported with an article review. 

### 2.1. Expert Selection

The experts were selected among those who have expertise in women’s and child’s health, nutrition, clinical psychology, and education. Contact details of these experts were obtained through various networks within the universities and professional associations. The experts were invited by telephone and e-mail to participate in a virtual meeting, which was audio-recorded. In total, 11 experts representing each field to have an effective discussion participated in the expert review meeting. 

### 2.2. Expert Meeting 

The methodology applied was a focus group discussion with experts. A virtual meeting was conducted to facilitate the discussion among experts. The questions included represented the situation on the available interventions to prevent IDA in Indonesia. The experts reviewed the national guidelines for IDA, explained the existing problems found in the daily practice, and proposed possible solutions for the problems. During the meeting, the opinion of the experts was asked regarding which factors contributed to the low consumption of iron tablets among women and toddlers and what the possible solutions could be to increase the iron intake. The discussion was conducted online, recorded, and then transcribed and analyzed descriptively. 

## 3. Results

A total of eleven experts attended the meeting: two males and nine females. The expert group consisted of two specialists in obstetrics and gynecology, one pediatrician specialized in nutrition and metabolism, two pediatricians specialized in growth and development, two specialists in clinical nutrition, one expert in community nutrition, one expert in clinical psychology, one expert from midwife’s association, and one expert from education. The experts have great experience in their expertise, ranging from 10 years to more than 30 years (Table 1).

## 4. Discussion

### 4.1. Indonesian Government’s Programs to Overcome Anemia 

The Government of Indonesia (GoI) has compiled various programs to control anemia (Table 2). These programs are part of the National Strategy to Accelerate Stunting Prevention 2018–2024. The targets of these programs are female adolescents, pregnant and lactating mothers, and also infants and young children. 

Although many programs have been run by the GoI, the effectiveness of these programs is still questioned since there has not been a comprehensive evaluation of the interventions. Eliminating stunting is the main priority of the GoI, and IDA is closely related to stunting. The listed interventions in Table 2 are those associated with reducing IDA.

### 4.2. Factors Affecting the Anemia Control Program 

The causes of anemia are multifactorial and complex. Poor intake of iron-rich foods was commonly blamed for being the main culprit. Nevertheless, there are many factors possibly affecting the success of a program to decrease anemia, such as government commitment to eliminate anemia, the capability of health providers, food availability, other micronutrients deficiencies which may contribute to erythrocyte production, and motivation of the patients themselves (Figure 1). 

Lack of political commitment and financial support decreases the efficiency of an anemia control program. From the supply side, lack of iron tablets in health facilities, lack of laboratory examinations, or training of health professionals affect the quality of service delivered [20]. On the other hand, the decision to take iron tablets is dependent on the patient’s adherence. Older women, higher educated women, working mothers, women who gave birth to a lower number of children, women with high socioeconomic status, and women whose husbands are present during prenatal visits are more likely to adhere to iron supplementation. Those who had a history of abortion, better knowledge of the symptoms and consequences of anemia, or received health education also had higher compliance [21,22,23]. Regardless of adherence problems, iron absorption in humans is dependent on the availability of iron in the diet. Poor tolerance leads to poor compliance to adhere to iron supplementation. Therefore, fortified food or beverage could be the solution to improve tolerance. Iron is mainly obtained from food. There are two primary forms of dietary iron: heme and non-heme iron. Heme iron is derived from animal sources, mainly meat, fish, and poultry, while non-heme iron is found in plant foods like whole grains, nuts, seeds, legumes, and leafy greens. Heme iron has a higher bioavailability and higher absorption to the body than non-heme iron. The absorption of non-heme iron is more sensitive to the presence of nutrients that may enhance the absorption, such as ascorbic acids, or that may inhibit the absorption, such as phytates and tannins [24]. Genetically modified plants enriched in iron with reduced phytate content could be beneficial. The Indonesian Food Consumption Survey of 2018 revealed that the consumption of heme iron was lower than the non-heme iron (32.2% vs. 67.8%, respectively).

### 4.3. Effects and Possible Interventions in Each Life Stage 

Unfortunately, the Indonesian government did not organize a follow-up of the different programs to assess their effectiveness. Understanding the existing anemia control program and factors affecting the effectiveness of the program, we should be aware that more interventions are necessary to combat those challenges. People with anemia may not have any symptoms, but if the condition is left untreated, it may cause many and long-lasting health issues. A life course approach to health is based on the understanding that interactions between environmental exposures and life experiences are cumulative over time. These interactions have an impact not only on the individual but also on the population and even the next generation. Figure 2 depicts the anemia effects throughout different life stages: early childhood, adolescence, adulthood, and during preconception/prenatal stage. 

Anemia during adolescence is quite common. IDA during adolescence may reduce physical work capacity and cognitive function. The problem with adolescents is that the anemia would not only affect their current health status but also continue to their next life stage. A young woman growing up with anemia would potentially still have anemia in adulthood. A pregnant woman with anemia has an increased risk to deliver preterm and develop pre-eclampsia and post-partum hemorrhage, increasing maternal mortality risk. Furthermore, the fetus is at increased risk for having growth retardation and infection. When born, this baby is at increased risk for impaired mental and psycho-motor development and growth delay. With the growing problem of anemia around the world, there is a risk for a generation with potential failure to thrive and development delay, which could ultimately result in a generation with low productivity. The existing interventions focused on the preconception/prenatal stage and early childhood (Figure 3). Interventions early in life will have a long-term effect. 

However, the efforts are still insufficient. More should be done in the life stage before entering the prenatal stage. Interventions should be performed in every life stage, and especially during early adolescence and in every life stage. Iron tablet supplementation for female teenagers had been promoted and implemented. Nevertheless, the compliance of taking iron tablets is still low because of adverse effects of iron intake and discontinuation during school holidays. Health education in schools and personal nutritional counseling for teenagers should increase awareness about anemia. Education should include information on iron-rich food, including enhancing and inhibiting factors affecting iron absorption. The experts proposed peer group counseling and boosting the health campaigns through influencers as effective action plans. These recommendations were based on the expertise of experts regarding previous studies, which found that most teenagers did not want to consume iron tablets because of the influence of their peers. 

## 5. Conclusions

All experts agreed that a life course approach should be used as a framework to develop interventions in preventing iron deficiency anemia in each life stage. The existing interventions have not fully addressed the needs. This expert opinion proposed that health education and nutritional intervention can start from adolescence. Although this is an Indonesian consensus, these findings and recommendations are worldwide applicable. Further research to explore the effectiveness of these interventions would be important.

## Figures and Tables

**Figure 1 nutrients-14-00277-f001:**
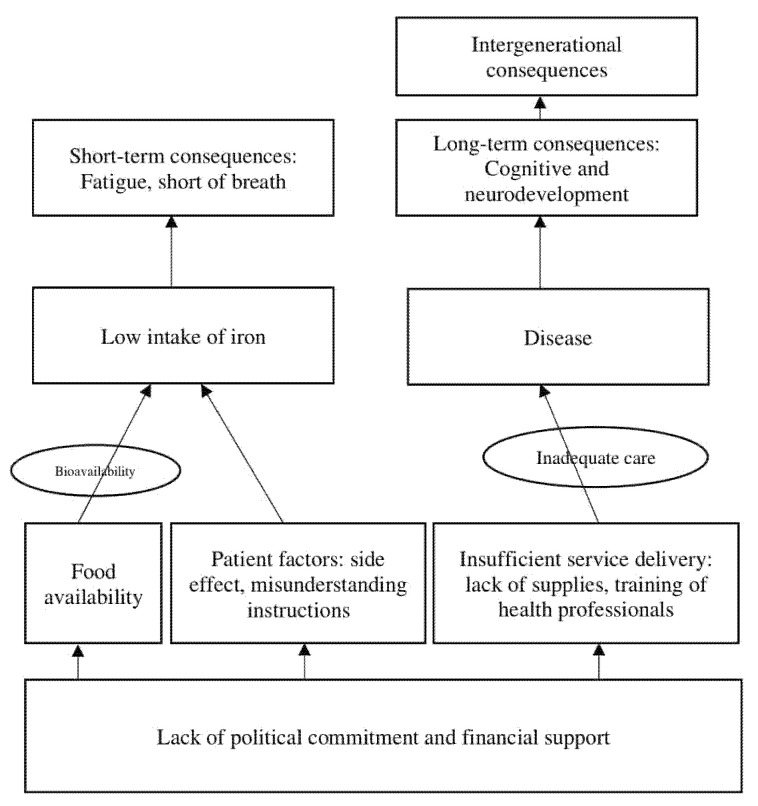
Various factors affecting the anemia control program (modified from UNICEF conceptual framework for under-nutrition) [19].

**Figure 2 nutrients-14-00277-f002:**
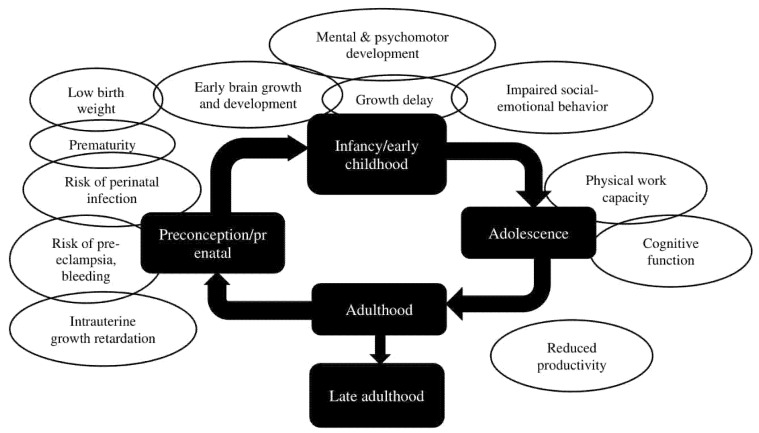
Effects of anemia throughout life stages [25,26,27].

**Figure 3 nutrients-14-00277-f003:**
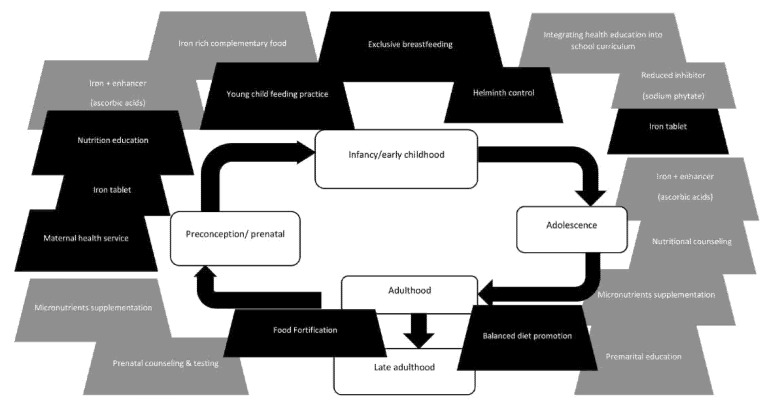
Possible interventions throughout life stages (black—existing interventions; grey—proposed interventions) [28,29,30,31,32].

**Table 1 nutrients-14-00277-t001:** Experts’ demographics.

Variable	Category	Frequency	Percentage
Gender	Male	2	18.1
	Female	9	81.8
Age	40–50 years old	4	36.3
	50–60 years old	5	45.4
	>60 years old	2	18.1
Expertise	Obstetrics and Gynecology	2	18.1
	Pediatrician specialized in Nutrition and Metabolic		
	Pediatrician specialized in Growth and Development	1	9.0
	Specialist in Clinical Nutrition		
	Community Nutrition	2	18.1
	Clinical Psychology	2	18.1
	Midwife	1	9.0
	Education	1	9.0
Experience	10–20 years	4	36.3
	20–30 years	5	45.4
	>30 years	2	18.1

**Table 2 nutrients-14-00277-t002:** Interventions to prevent anemia in the National Strategy.

No	Interventions	Population Target
1	Providing iron tablets and folic acid	Female adolescents, pregnant mothers
2	Early breastfeeding initiation and promotion of exclusive breastfeeding up to 6 months of age [17]	Lactating mothers, infants
3	Providing information on infant and young child feeding practices	Infant and young child
4	*Helminth* prevention and control	Under-five and school-age children
5	Improving maternal healthcare services	Pregnant mothers
6	Nutrition education for pregnant mothers	Pregnant mothers
7	Balanced diet promotion (including recommending legumes consumption) [18]	General population
8	Food fortification (including the possibility of genetically modified plants)	General population

## Data Availability

Not applicable.

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
