# Peer review of "A Life Course Approach to the Prevention of Iron Deficiency Anemia in Indonesia"

_nutrients, 2022, doi:10.3390/nu14020277_

Round 1

Reviewer 1 Report

This is an interesting paper on the hudge health problem of iron deficiency in Indonesia, applicable also to other developing countries. It deserves some comments

Comments

  1. Poor meat, fish or poultry availability is probably the main cause of iron deficiency in developing countries. The authors should more insist on this point, especially when they mention iron-rich foods.
  2. As the authors mentioned, plants non-heme iron is much less absorbed than heme iron. However, iron content of legumes is much higher than that of cereals or green vegetables (except spinach). When meat is not available, encouraging legumes consumption could be a solution to increase iron intakes. Do the authors know whether this solution was suggested in the interventions to prevent IDA (Table 2) ?
  3. Another solution to enhance non-heme iron absorption in countries with poor meat availability is to genetically modify plants rich in iron in order to reduce phytates content. Was this possibility suggested ?
  4. Difficulties to adhere to iron supplementation is mentioned. Was the poor tolerance of the supplementation one of the reason and information to improve tolerance was it provided ?
  5. Table 2 : promotion of exclusive and prolonged breastfeeding is obviously an important challenge in developing countries, especially to prevent infants against infections. However, since iron content of breast milk is very low, I do not understand why promotion of breastfeeding was included into strategies to prevent iron deficiency.
  6. Table 2 : more precisions about information on infant and young child feeding practices (item n° 3) and balanced diet (item n° 7) should be provided by the authors.
  7. Page 4, line 131 : the authors wrote that non-heme iron is found in animal-based foods. What are those foods ?
  8. Figure 3 is difficult to read. Could it be improved ?

Author Response

Dear reviewer

We thank you for your constructive suggestions to improve our manuscript " A Life Course Approach to the Prevention of Iron Deficiency Anemia in Indonesia". All your comments have been  considered and are listed in the manuscript and the table below.

Poor meat, fish or poultry availability is probably the main cause of iron deficiency in developing countries. The authors should more insist on this point, especially when they mention iron-rich foods.

We have added this notion in the introduction section. " Poor meat, fish or poultry availability is likely to be one of the main causes of iron deficiency in developing countries"

As the authors mentioned, plants non-heme iron is much less absorbed than heme iron. However, iron content of legumes is much higher than that of cereals or green vegetables (except spinach). When meat is not available, encouraging legumes consumption could be a solution to increase iron intakes. Do the authors know whether this solution was suggested in the interventions to prevent IDA (Table 2) ?

We agree with the reviewer and adapted Table 2: "Balanced diet promotion (including recommending legumes consumption)"

Another solution to enhance non-heme iron absorption in countries with poor meat availability is to genetically modify plants rich in iron in order to reduce phytates content. Was this possibility suggested ?

Again, we agree with the reviewer and added this information in Table 2.

"Food fortification (including the possibility of genetically modified plants)"

We also added the statement in the discussion section: "Genetically modified plants enriched in iron with reduced phytate content could be beneficial."

Difficulties to adhere to iron supplementation is mentioned. Was the poor tolerance of the supplementation one of the reason and information to improve tolerance was it provided ?

We added the following:  “Poor tolerance leads to poor compliance to adhere iron supplementation.  Therefore fortified food or beverage could be the solution to improve the tolerance”

Table 2 : promotion of exclusive and prolonged breastfeeding is obviously an important challenge in developing countries, especially to prevent infants against infections. However, since iron content of breast milk is very low, I do not understand why promotion of breastfeeding was included into strategies to prevent iron deficiency.

We adapted the table. Many infants < 6 months are not exclusively breastfed and receive unmodified cow's milk (not infant formula). Therefore, exclusive breastfeeding UP TO 6 months is recommended.

The table reads now: "Early breastfeeding initiation and promotion of exclusive breastfeeding up to 6 months of age".

Table 2 : more precisions about information on infant and young child feeding practices (item n° 3) and balanced diet (item n° 7) should be provided by the authors.

Two references were added

1. Unit Kerja Koordinasi Nutrisi dan Penyakit Metabolik PP IDAI. Rekomendasi Praktik Pemberian Makanan Berbasis Bukti pada Bayi dan Batita di Indonesia untuk Mencegah Malnutrisi. 2015

2.           Kemenkes RI. Pedoman Gizi Seimbang. Peraturan Menteri Kesehatan RI No. 41/2014.

Page 4, line 131 : the authors wrote that non-heme iron is found in animal-based foods. What are those foods ?

We thank the reviewer to have noticed the typo (the "non" disappeared, it was "non-animal-based foods"). We changed the sentence. It reads now: "Non-heme iron is found in plant foods like whole grains, nuts, seeds, legumes, and leafy greens."

Fig 3 is difficult to read. Could it be improved ?

Done

Reviewer 2 Report

The article by Sunghar et al. was very easy and concise to read, following an expert panel discussion methodology on iron deficiency in the Indonesian population. I do not have many comments about this paper, the introduction is very comprehensive and well understood.

  • I don't know if the review description (line 69) would be the most appropriate in this paper.
  • Perhaps the methodology used by the authors is focus group or discussion groups, I suggest that you look for this type of methodology and include it in material and methods.
  • A detailed list of the questions that were asked in the discussion as well as how the meeting was conducted and who wrote the answers/responses would be necessary to report.
  • Table 1 seems to be unordered (variable frequency in expert category).
  • In the discussion it would be nice to find data on the different programs increasing or decreasing IDA or anemia in Indonesia to assess the effectiveness of these programs.
  • Also in the discussion, if the authors could propose concrete action plans to improve the programs, adapted to the population in Indonesia.
  • The resolution of the figures is not adequate and makes it difficult to understand. 

Other minor comments could be:
- Affiliation "10" is not well indexed.
- Line 124, what is lower birth order?

Author Response

Dear reviewer

We thank you for your constructive suggestions to improve our manuscript " A Life Course Approach to the Prevention of Iron Deficiency Anemia in Indonesia". All your comments have been  considered and are listed in the manuscript and the table below.

The article by Sunghar et al. was very easy and concise to read, following an expert panel discussion methodology on iron deficiency in the Indonesian population. I do not have many comments about this paper, the introduction is very comprehensive and well understood.

We thank the reviewer for the nice comment

I don't know if the review description (line 69) would be the most appropriate in this paper.

We rephrased the sentence: "Since the impact of iron deficiency is long term and affects each life stage, it is important to consider different life stages in the prevention of IDA"

Perhaps the methodology used by the authors is focus group or discussion groups, I suggest that you look for this type of methodology and include it in material and methods.

The methodology applied was a focus group discussion with experts. (added to manuscript)

A detailed list of the questions that were asked in the discussion as well as how the meeting was conducted and who wrote the answers/responses would be necessary to report.

During the meeting, the opinion of the experts was asked regarding which factors contributed to the low consumption of iron tablets among women and toddlers, and what the possible solutions could be to increase the iron intake. The discussion was conducted online, recorded, and then transcribed and analyzed descriptively. (added to manuscript)

Table 1 seems to be unordered (variable frequency in expert category).

We thank the reviewer to have noticed these typo's.

In the discussion it would be nice to find data on the different programs increasing or decreasing IDA or anemia in Indonesia to assess the effectiveness of these programs.

We added to the manuscript: "Unfortunately, the government did not organize a follow-up of the different programs to assess their effectiveness".  

Also in the discussion, if the authors could propose concrete action plans to improve the programs, adapted to the population in Indonesia.

We added the following in the discussion section: "The experts proposed peer group counseling and boosting the health campaigns through influencers as effective action plans.  These recommendations were based on the expertise of experts regarding previous studies which found that most teenagers did not want to consume iron tablets because of the influence of their peers."

The resolution of the figures is not adequate and makes it difficult to understand.

Done

Affiliation "10" is not well indexed.

Thanks; has been changed

Line 124, what is lower birth order?

We changed to " women who gave birth to a lower number of children"